# Load-Swing Attenuation in a Quadcopter–Payload System Through Trajectory Optimisation

**DOI:** 10.3390/s25175518

**Published:** 2025-09-04

**Authors:** Barry Feng, Arash Khatamianfar

**Affiliations:** School of Electrical Engineering and Telecommunications, UNSW Sydney, Sydney 2052, Australia; barry.feng@unsw.edu.au

**Keywords:** quadcopter, load-swing attenuation, time-varying LQR, trajectory optimization, CUDA-accelerated tag detection, visual servoing

## Abstract

Advancements in multi-rotor quadcopter technology and sensing capabilities have led to their increased utilisation for last-mile delivery. However, battery capacity constraints limit their use in extended-distance delivery scenarios. A visual servoing implementation is first proposed that leverages a CUDA-accelerated tag detection algorithm for real-time pose estimation of the target. A new approach is then developed to enhance quadcopter package collection by implementing a control scheme to attenuate aggressive load-swing in a payload arm that shifts from horizontal to vertical after obtaining a vertically mounted payload. The motion of the payload arm imposes a shift in the system’s centre of mass, leading to a possible instability. A non-linear control scheme is then introduced to address this problem through attenuation of the residual energy from payload oscillation. The performance of the visual servoing approach is validated through both numerical simulations and a physical quadcopter implementation, along with the performance of the load-swing attenuation through numerical simulations.

## 1. Introduction

The advent of unmanned aerial vehicles (UAVs) has opened up substantial possibilities for payload delivery applications [1]. Nonetheless, the maximum operational range of payload delivery drones is limited to the flight-time endurance of a single multirotor. Moreover, the endurance per unit payload of a single multirotor can be decomposed into a function of the multirotor’s mass and electrical power requirements. This implies that an increase in the endurance per unit payload inherently leads to an increase in drone mass and power requirements [2]. Traditional package delivery networks have been under pressure due to recent surges in e-commerce usage, with specific congestion points appearing at the last-mile stage. A method to minimise this congestion in urban environments is the transfer of payloads from buildings to multirotors. To avoid landing structures outside the buildings, this payload transfer process would necessarily require the multirotor to retrieve the payload away from its centre of mass through an extension arm. This induces significant shifts in the centre of mass following payload retrieval and applies unwanted forces on the multirotor, potentially leading to flight instability. This article proposes a transformation in multirotor dynamics by allowing the payload arm to swing down from the initial horizontal position. To minimise the impact of the load swing during this transformation, a set of control laws is introduced to attenuate the swing motion. Although substantial research has been conducted on controlling slung payload systems in trajectory tracking applications [3,4,5], this article emphasises attenuating the load swing from the worst-case horizontal initial condition.

### 1.1. Related Work

The analysis of the rigid-body dynamics of slung-load systems originated from their application in multi-lift helicopters, where multiple helicopters were arranged to transport a single payload suspended by cables between them [6]. Although these derivations do not directly translate to the conditions encountered in quadcopter situations, as the relative motion between each helicopter is constrained in a fixed quadcopter structure, the study showcased the viability of payload conveyance by multi-lift configurations. Bisgaard et al. extended this study by presenting generic equations of motion for slung-payload systems with any number of carrier aircraft, taking into account the impacts of wire tension and quadratic aerodynamic drag [7]. Ref. [8] extended this by deriving a similar model through the Euler–Lagrange method.

The process of quadcopters approaching visual targets has been explored extensively in various contexts, including quadcopter docking and autonomous landing. Shankar et al. presented a pose-based visual servoing approach using monocular vision to establish the pose of an approaching drone relative to the extended arm of a hovering drone [9]. In combination with the PBVS implementation, Shanker et al. [9] used active, illuminated markers to speed up target identification by reducing camera exposure. Keipour et al. presented an image-based visual servoing approach to demonstrate tracking a landing target, where the pixel difference between the extracted target features in the quadcopter’s real-time view and a pre-defined view of the target was used as the error signal [10]. While image-based visual servoing is more insensitive to calibration errors, Keshmiri [11] reveals that IBVS schemes may not always guarantee completion due to the existence of singularities in the interaction matrix.

The process of load-swing attenuation while transporting multirotor-carried payloads has been widely researched in the literature. However, a number of these approaches cater to small-scale load-swing angles [5,12,13] where the multirotor dynamics can largely be linearised. In practice, this approach is infeasible for attenuation of the initial response of a payload transfer system. Classical control methods such as proportional–integral–derivative (PID) and its variants treat the payload as a disturbance [14] and aim to minimise the swing angle. Us et al. [13] extended this through optimal control by designing an LQR controller for trajectory tracking with a slung load using Jacobian linearisation around the pendulum’s downward position. While this approach reduced overshoot compared to PD control, it suffered from a loss of accuracy at large swing angles due to linearisation. Alothman et al. [15] then extended this with an iterative LQR (iLQR) scheme to allow for the controller to respond to changes in operating point induced by the slung payload. Bangura et al. [16] demonstrated a linear constrained MPC scheme where linearised quadcopter dynamics were optimised through a quadratic programming problem, allowing flight tests to show minor tracking errors at low velocities and a 50% reduction in swing angle. To improve the controller’s response to model uncertainty, Yuan et al. [17] presented an H-∞ controller that was optimised using linear matrix inequalities to achieve stable outputs for various disturbances.

In order to improve controller response to the non-linearities of the quadcopter–payload system, Guerrero-Sanchez et al. [18] proposed both a PD controller with a nonlinear coupling term and a fully nonlinear control law derived via Lyapunov stability analysis. Other nonlinear techniques, such as nonlinear MPC [19] and partial feedback linearisation [20], have also been explored for the swung-load problem. Model-free approaches, such as reinforcement learning methods, have also been presented to bypass the difficulties in system identification and nonlinear analysis. Faust et al. [21] and Panetsos et al. [22], overcame model inaccuracies by learning system parameters iteratively to improve swing attenuation for large angles.

### 1.2. Contributions

The state of the art is extended by presenting a feasible control solution for quadcopter flight while carrying a swung payload from an initially horizontal position. This extensively differs from existing literature by presenting a feasible and optimal solution to generate the shortest path from a worst-case initial swing-angle to the downward equilibrium condition. In existing literature, the techniques that employ a single linearisation in the downward payload position no longer produce accurate approximations when the payload begins in a horizontal position. In addition, they do not offer solutions with notions of optimality in terms of the shortest attenuation path. In this article, an application of pose-based visual servoing is then presented using time-invariant LQR (TILQR) feedback and a custom CUDA [23] port of the AprilTag detection library [24] to enable the quadcopter to approach the payload. The CUDA implementation replaces the thresholding, segmentation and clustering steps to yield an image-to-pose latency improvement by more than three times compared to public CPU implementations on similar grade hardware. An application of constrained trajectory optimisation is then used to generate a feasible swing-attenuating path for the quadcopter after obtaining the slung payload. The stability of this flight path is improved under local feedback through a time-varying linear quadratic regulator (TVLQR), augmented with an inverse-dynamics feedforward control strategy. The feasibility, controllability and observability of the time-varying closed-loop system are then evaluated to ensure the trajectory-tracking action is able to be implemented. Numerical simulations are first presented to demonstrate the effectiveness of visual servoing, followed by flight tests with an Alien 450 quadcopter. The effectiveness of the swing-attenuating trajectory and feedback controller is demonstrated through numerical simulations in the presence of model uncertainties up to ±20% variations in payload and quadcopter mass, as well as the incorporation of wind disturbances that are modelled as force vectors with a Gaussian distributed magnitude.

## 2. System Model

### 2.1. Model Assumptions

The quadcopter–payload dynamic model can be derived using the assumptions made in [8], while the model is constrained such that the quadcopter structure abides by rigid body dynamics. The quadcopter is considered symmetric across the *x* and *y* axes if all joints are allowed to settle to a steady state condition. This enables the centre of gravity of the quadcopter to be placed at the origin of the body-fixed frame of the quadcopter and the quadcopter rotors to behave as ideal actuator disks. The effect of the payload on the quadcopter’s dynamics is modelled under the assumption that the load behaves as a point mass on the end of a spherical pendulum. The spherical pendulum is then assumed to be mechanically connected to the centre of mass of the quadcopter structure. In addition, the mass of the pendulum arm is considered negligible while the pendulum arm is infinitely stiff, and the suspension point is assumed frictionless, allowing the pendulum arm motion to behave as an ideal spherical pendulum.

### 2.2. Swung Payload Equations of Motion

The dynamic model of the quadcopter with an attached slung payload is described by Equations (1)–(8). The reference frame in which the model was derived is shown by Frame W in Figure 1. The end of the pendulum is represented by a point P of length *l*. Full details of the use of the remaining reference frames in the model derivation are provided in Section A.1.(1)$$\begin{array}{c} \begin{array}{cc} F_{x_{q}} & =\left(m_{l}+m_{q}\right)\ddot{x_{q}}+lm_{l}\cos\left(\phi_{l}\right)\cos\left(\theta_{l}\right)\ddot{\theta_{l}}-lm_{l}\sin\left(\phi_{l}\right)\sin\left(\theta_{l}\right)\ddot{\phi_{l}}\\ \; & \;-lm_{l}\cos\left(\phi_{l}\right)\sin\left(\theta_{l}\right){\dot{\phi_{l}}}^{2}-lm_{l}\cos\left(\phi_{l}\right)\sin\left(\theta_{l}\right){\dot{\theta_{l}}}^{2}\\ \; & \;-2lm_{l}\cos\left(\theta_{l}\right)\sin\left(\phi_{l}\right)\dot{\theta_{l}}\dot{\phi_{l}}+k_{q,d,x}\dot{x_{q}}\;, \end{array} \end{array}$$(2)$$\begin{array}{c} \begin{array}{cc} F_{y_{q}} & =\left(m_{l}+m_{q}\right)\ddot{y_{q}}-lm_{l}\sin\left(\phi_{l}\right){\dot{\phi_{l}}}^{2}+lm_{l}\cos\left(\phi_{l}\right)\ddot{\phi_{l}}+k_{q,d,y}\dot{y_{q}}\;, \end{array} \end{array}$$(3)$$\begin{array}{c} \begin{array}{cc} F_{z_{q}} & =\left(m_{l}+m_{q}\right)\ddot{z_{q}}+g\left(m_{l}+m_{q}\right)+lm_{l}\cos\left(\theta_{l}\right)\sin\left(\phi_{l}\right)\ddot{\phi_{l}}+k_{q,d,z}\dot{z_{q}}\\ \; & \;+lm_{l}\cos\left(\phi_{l}\right)\sin\left(\theta_{l}\right)\ddot{\theta_{l}}+lm_{l}\cos\left(\phi_{l}\right)\cos\left(\theta_{l}\right){\dot{\phi_{l}}}^{2}\\ \; & \;+lm_{l}\cos\left(\phi_{l}\right)\cos\left(\theta_{l}\right){\dot{\theta_{l}}}^{2}-2lm_{l}\sin\left(\phi_{l}\right)\sin\left(\theta_{l}\right)\dot{\theta_{l}}\dot{\phi_{l}}\;, \end{array} \end{array}$$(4)$$\begin{array}{c} \begin{array}{cc} \tau_{\phi_{q}} & =I_{\mathrm{xx}}\ddot{\phi_{q}}\;, \end{array} \end{array}$$(5)$$\begin{array}{c} \begin{array}{cc} \tau_{\theta_{q}} & =I_{\mathrm{yy}}\ddot{\theta_{q}}\;, \end{array} \end{array}$$(6)$$\begin{array}{c} \begin{array}{cc} \tau_{\psi_{q}} & =I_{\mathrm{zz}}\ddot{\psi_{q}}\;, \end{array} \end{array}$$(7)$$\begin{array}{c} \begin{array}{cc} 0 & =l^{2}m_{l}\ddot{\phi_{l}}+lm_{l}\cos\left(\phi_{l}\right)\ddot{y_{q}}+lm_{l}\cos\left(\theta_{l}\right)\sin\left(\phi_{l}\right)\ddot{z_{q}}\\ \; & \;-lm_{l}\sin\left(\phi_{l}\right)\sin\left(\theta_{l}\right)\ddot{x_{q}}-glm_{l}\cos\left(\theta_{l}\right)\sin\left(\phi_{l}\right)\\ \; & \;+l^{2}m_{l}\cos\left(\phi_{l}\right)\sin\left(\phi_{l}\right){\dot{\theta_{l}}}^{2}+k_{l,d}l^{3}{\dot{\phi_{l}}}^{2}\frac{\dot{\phi_{l}}}{|\dot{\phi_{l}}|}\;, \end{array} \end{array}$$(8)$$\begin{array}{c} \begin{array}{cc} 0 & =lm_{l}\cos\left(\phi_{l}\right)\left(-g\sin\left(\theta_{l}\right)+\cos\left(\theta_{l}\right)\ddot{x_{q}}+\sin\left(\theta_{l}\right)\ddot{z_{q}}+l\cos\left(\phi_{l}\right)\ddot{\theta_{l}}\right)\\ \; & \;-2l^{2}m_{l}\sin\left(\phi_{l}\right)\cos\left(\phi_{l}\right)\dot{\theta_{l}}\dot{\phi_{l}}+k_{l,d}l^{3}{\dot{\theta_{l}}}^{2}\frac{\dot{\theta_{l}}}{|\dot{\theta_{l}}|}\;, \end{array} \end{array}$$
where $F_{x_{q}}$, $F_{y_{q}}$, and $F_{z_{q}}$ represent the driving forces on the quadcopter in the *x*, *y*, and *z* directions within frame **W**, respectively. $\tau_{\phi_{q}}$, $\tau_{\theta_{q}}$, and $\tau_{\psi_{q}}$ represent the torque applied to the quadcopter roll, pitch, and yaw orientations, respectively. Equations (7) and (8) represent the swing dynamics of the system and are seen to have no external force. This reveals the under-actuated behaviour of the swung load. The equations of motion can then be simplified to matrix form for entry into numerical solvers:(9)$$\begin{array}{c} M\left(q\right)\ddot{q}+C\left(\dot{q},q\right)\dot{q}+G\left(q\right)+D\left(\dot{q}\right)=Q\;, \end{array}$$
where $M\in R^{8\times 8}$ represents the symmetric mass matrix, $C\in R^{8\times 8}$ represents the Coriolis matrix, $G\in R^{8}$ represents the gravity matrix, and $Q\in R^{8}$ contains the external driving forces on the system for all *q*. The generalised coordinate *q* and its time-derivative $\dot{q}$ are defined as follows:(10)$$\begin{array}{cc} q & ={\left[\begin{array}{cccccccc} x_{q} & y_{q} & z_{q} & \phi_{q} & \theta_{q} & \psi_{q} & \phi_{l} & \theta_{l} \end{array}\right]}^{T}\;, \end{array}$$(11)$$\begin{array}{cc} \dot{q} & ={\left[\begin{array}{cccccccc} \dot{x_{q}} & \dot{y_{q}} & \dot{z_{q}} & \dot{\phi_{q}} & \dot{\theta_{q}} & \dot{\psi_{q}} & \dot{\phi_{l}} & \dot{\theta_{l}} \end{array}\right]}^{T}\;. \end{array}$$

A detailed derivation of the equations of motion is included in Section A.1, and the complete matrices M,C,G,D,Q are included in Section A.2.

### 2.3. Fixed-Arm Equations of Motion

For the rendezvous case, the equations of motion can be simplified by substituting $\phi_{l}\;,{\dot{\phi }}_{l}\;,{\ddot{\phi }}_{l}=0$ and $\theta_{l}=\frac{\pi }{2}\;,{\dot{\theta }}_{l}\;,{\ddot{\theta }}_{l}=0$ to indicate the payload arm in a fixed position. This yields(12)$$\begin{array}{cc} F_{x_{q}} & =\left(m_{l}+m_{q}\right)\ddot{x_{q}}+k_{q,d,x}\dot{x_{q}}\;, \end{array}$$(13)$$\begin{array}{cc} F_{y_{q}} & =\left(m_{l}+m_{q}\right)\ddot{y_{q}}+k_{q,d,y}\dot{y_{q}}\;, \end{array}$$(14)$$\begin{array}{cc} F_{z_{q}} & =\left(m_{l}+m_{q}\right)\ddot{z_{q}}+g\left(m_{l}+m_{q}\right)+k_{q,d,z}\dot{z_{q}}\;, \end{array}$$(15)$$\begin{array}{cc} \tau_{\phi_{q}} & =I_{\mathrm{xx}}\ddot{\phi_{q}}\;, \end{array}$$(16)$$\begin{array}{cc} \tau_{\theta_{q}} & =I_{\mathrm{yy}}\ddot{\theta_{q}}\;, \end{array}$$(17)$$\begin{array}{cc} \tau_{\psi_{q}} & =I_{\mathrm{zz}}\ddot{\psi_{q}}\;, \end{array}$$(18)$$\begin{array}{cc} \tau_{\phi_{l}} & =0\;, \end{array}$$(19)$$\begin{array}{cc} \tau_{\theta_{l}} & =0\;. \end{array}$$

In state-space representation, this becomes(20)$$\begin{array}{ccc} \; & \dot{x} & =Ax+Bu+f \end{array}$$
where,$$\begin{array}{cc} x & ={\left[x_{q},y_{q},z_{q},\phi_{q},\theta_{q},\psi_{q},\phi_{l},\theta_{l},{\dot{x}}_{q},{\dot{y}}_{q},{\dot{z}}_{q},\dot{\phi_{q}},\dot{\theta_{q}},\dot{\psi_{q}},\dot{\phi_{l}},\dot{\theta_{l}}\right]}^{T}\\ u & ={\left[F_{x_{q}},F_{y_{q}},F_{z_{q}},\tau_{\phi_{q}},\tau_{\theta_{l}},\tau_{\psi_{l}},0,0\right]}^{T}\\ f & ={\left[0_{1\times 8},-g,0_{1\times 5}\right]}^{T}\\ A & =\left[\begin{array}{cc} 0_{6\times 6} & \mathrm{diag}\left(1_{1\times 6}\right)\\ 0_{6\times 6} & \mathrm{diag}\left(-\frac{k_{q,d,x}}{m_{l}+m_{q}},-\frac{k_{q,d,y}}{m_{l}+m_{q}},-\frac{k_{q,d,z}}{m_{l}+m_{q}},0_{1\times 3}\right) \end{array}\right]\;,\\ B & =\left[\begin{array}{c} 0_{8\times 8}\\ \mathrm{diag}\left(\frac{1}{m_{l}+m_{q}},\frac{1}{m_{l}+m_{q}},\frac{1}{m_{l}+m_{q}},\frac{1}{i_{xx}},\frac{1}{i_{yy}},\frac{1}{i_{zz}},0,0\right) \end{array}\right]\;. \end{array}$$

### 2.4. Trajectory Optimisation Scheme

In order to constrain the flight path of the quadcopter during the swing attenuation process, an open-loop trajectory is computed through direct collocation. This trajectory optimisation process creates a trajectory that theoretically enables the system to reach the control objective through the state space [25]. While it is also known that this technique suffers from returning suboptimal solutions due to convergence to local minima, convergence to the same solution across a range of problem initialisations can still be obtained. Furthermore, the reference frame for the trajectory is always considered to be referenced to the parcel itself. Therefore, the initial pose of the system is always taken as (0, 0, 0) m.

While the search for a policy that allows an optimal trajectory to be generated from any location in the state space of a closed-loop solution is desirable, the computation of optimal solutions for all points in the state space has exponential time complexity and thus becomes intractable for higher-dimensional problems such as the swung-payload system. As such, an open-loop solution is generated using trajectory optimisation to create a trajectory that allows the system to reach stability through the state space [25]. However, the definition of a single trajectory means that stability is undefined outside the bounds of the trajectory and, thus, a trajectory-tracking controller must also be implemented to ensure the system follows the optimal trajectory [26]. For this article, the continuous-time problem in Equation (21) is formulated as a discrete nonlinear programming (NLP) problem in Equation (22), now subject to constraints expressed in terms of the polynomial interpolation of the states and control variables:(21)$$\begin{array}{cc} \min_{u(t),x(t)}\; & \int_{0}^{T}u^{2}\left(\tau \right)d\tau \\ s.t.\; & \dot{x}=f\left(t,x\left(t\right),u\left(t\right)\right)\\ \; & x_{\min}\leq x\leq x_{\max}\\ \; & u_{\min}\leq u\leq u_{\max} \end{array}$$(22)$$\begin{array}{cc} \min_{\begin{array}{c} u_{0}\ldots u_{N}\\ x_{0}\ldots x_{N} \end{array}}\; & \sum_{k=0}^{N-1}\frac{t_{k+1}-t_{k}}{2}\left(u_{k}^{2}+u_{k+1}^{2}\right)\\ s.t.\; & x_{k+1}-x_{k}=\frac{t_{k+1}-t_{k}}{2}\left(f_{k}^{2}+f_{k+1}^{2}\right)\\ \; & x_{\min}\leq x_{k}\leq x_{\max}\\ \; & u_{\min}\leq u_{k}\leq u_{\max}\;, \end{array}$$
where u represents the quadcopter control forces, *f* represents the non-linear system dynamics, *k* represents the discrete timestep, and x represents the system state. The constrained NLP problem is then iteratively minimised using the PSOPT library [27]. The optimisation approach generates an optimal trajectory given a bounded initial state $x_{i}$, path $x_{n}$, and final state $x_{f}$. The selected trajectory input constraints are listed in Table 1, and the trajectory state constraints are listed in Table 2. The bounds placed on the quadcopter displacement states $x_{q},y_{q},z_{q}$ are selected considering the maximum flyable area within the testing environment. By extension, the bounds placed on the angular velocity of the quadcopter are selected using the maximum possible angular velocity obtained during manual flight. As the load is unactuated and the swing behaviour of the load is directly related to the quadcopter body acceleration, the minimum and maximum allowable angular velocities of the load’s angular velocity are left unconstrained:

The optimised trajectory in Figure 2 is then produced following the optimisation process. For conciseness, only the relevant position and derivative states are shown. To determine the discretisation error in the trajectory, the error between the optimised trajectory and system dynamics within the collocation points is explored using a method adopted from [25]. The unknown true trajectory solution, $\hat{x}$, $\hat{u}$, satisfies the system dynamics $\hat{\dot{x}}\left(t\right)=f\left(t,\hat{x},\hat{u}\right)$. The error between the optimised trajectory solution x,u, and the behaviour modelled by the true system dynamics can then be expressed as follows:(23)$$\begin{array}{cc} \epsilon (t) & =\dot{x}\left(t\right)-f\left(t,x,u\right)\;. \end{array}$$

The error term e(t) is expected to equal zero at each collocation point and deviate elsewhere. Therefore, increasing collocation points or using a higher-order polynomial interpolation yields discretised system dynamics that better resemble the true system dynamics. The relative error between each set of collocation points is then determined by defining the normalised absolute local error:(24)$$\begin{array}{c} \epsilon_{k}=\max_{i}\frac{\int_{t_{k}}^{t_{k+1}}\left|\epsilon_{i}\left(t\right)\right|dt}{w_{i}+1}\;,\;\;w_{i}=\max\left|x\right|\;. \end{array}$$

The optimised trajectory shown in Figure 2 with 64 knot points yields sufficiently small collocation errors to proceed to controller design.

## 3. Controller Design

### 3.1. Visual Servoing Controller

A pose-based visual servoing scheme is developed to accomplish the target-approaching task. The pose estimation is achieved using a CUDA-accelerated AprilTag [24] detection implementation, which utilises an onboard GPU to accelerate AprilTag marker detection. The GPU processes grey-scale images through an adaptive thresholding scheme and identifies potential tag regions through connected component analysis. By then grouping components through connected-component labelling and applying quadrilateral fitting to the clusters, the algorithm confirms the presence and image coordinates of AprilTags. A more detailed explanation of the CUDA implementation is provided in Section B.1. The publicly available CPU-based AprilTag implementation is then leveraged to apply tag decoding and homography-based pose estimation to identify the 3D pose of the payload relative to the quadcopter. A GPU-based detection scheme is chosen to minimise the latency between the quadcopter motion and the vision pose estimate.

#### 3.1.1. AprilTag Detection Performance

Table 3 provides a comparison between the performance of the CPU-based AprilTag3 v3.4.4 versus AprilTag CUDA v1.0. The CPU-based AprilTag3 v3.4.4 is profiled on both the Intel NUC and Jetson Orin Nano platforms using four threads with no decimation and a maximum Hamming bit error of two. AprilTag CUDA v1.0 is profiled with identical decimation and error parameters. For the CPU-based quadrilateral decoding and pose estimation stages following CUDA parsing, single-threaded operation is used to optimise performance as the low quadrilateral count makes multi-threading unnecessary. The executable is run with 10,000 iterations at each resolution to determine the execution time average (μ) and standard deviation (σ) over detections.

It can be seen from the results in Table 3 that at low resolutions (640 × 480) there is a small improvement in using AprilTag CUDA v1.0 compared to AprilTag3 v3.4.4. However, at the desired (1280 × 720) resolution, an astounding 322% improvement in the GPU implementation via AprilTag CUDA v1.0 can be observed. These results are similarly observed at the 1600 × 900 resolution (264% improvement). Furthermore, offloading the detection task to the GPU minimises the effects of the OS scheduler, as seen by the large standard deviation values when executing on the CPU.

#### 3.1.2. Control Strategy

To maintain the AprilTag within the field-of-view of the camera at all times and to prevent loss of localisation, a piecewise linear trajectory between the current pose and the target pose to be followed by the tracking controller is generated. This yields a two-step trajectory generated by Algorithm 1 to approach the target. The trajectory achieves the objective of maintaining the AprilTag within the camera field of view by first centring the target in the image frame, servoing along the *y* and *z* axes in the quadcopter body frame B in Figure 1. Upon centring the tag within axial bounds set by $\epsilon_{\mathrm{yz}}$, the quadcopter approaches the tag at a constant velocity until a final distance to the tag is reached. A linear quadratic regulator (LQR) is then used to track the piecewise linear trajectory. The overall structure of the controller combined with the reference trajectory generator is illustrated in Figure 3, where $q_{m}$ and ${\ddot{q}}_{\mathrm{IMU}}$ are defined as the pose estimate and acceleration states from the quadcopter system, and $q_{\mathrm{est}}$ and ${\dot{q}}_{\mathrm{est}}$ represent the filtered position and velocity. The tracking error between the reference state and the current state is defined as $\bar{x}=x_{0}-x$.
**Algorithm 1** Approach Trajectory. $x_{0}$, $y_{0}$ and $z_{0}$ represent the initial position of the quadcopter relative to the target, and $t_{f}$ represents the trajectory length in seconds. 1: **procedure** GenerateTrajectory($x_{0},y_{0},z_{0},t_{f}$) 2:    $t_{hold}\leftarrow 5.0\;s$; $T_{x\_start}\leftarrow 15.0\;s$; $T_{x\_end}\leftarrow 30.0\;s$; $z_{target}\leftarrow 0.6\;m$; $f_{s}\leftarrow 100$ Hz 3:    $\epsilon_{yz}\leftarrow 0.03\;m$▹ Maximum $\epsilon_{yz}$ chosen to be smaller than tag size 4:    $T_{max}\leftarrow \max\left(t_{f},T_{x\_end}\right)$ 5:    $N\leftarrow \lceil T_{max}\;\cdot \;f_{s}\rceil$ 6:    $t\leftarrow [0,1/f_{s},2/f_{s},\ldots ,T_{max}]$▹ Time vector with *N* samples 7:    $N_{hold}\leftarrow f_{s}\;\cdot \;T_{hold}$; $N_{x\_start}\leftarrow f_{s}\;\cdot \;T_{x\_start}$; $N_{x\_end}\leftarrow f_{s}\;\cdot \;T_{x\_end}$ 8: 9:    **Forward Trajectory:**▹ Approach target in x-axis at $t=T_{x\_start}$ if target centred10:    $x\left[i\right]\leftarrow \left\{\begin{array}{cc} x_{0} & i<N_{x\_start}\\ x_{0}+\alpha \left(i\right)\;\cdot \;\frac{i-N_{x\_start}}{N_{x\_end}-N_{x\_start}}\left(x_{target}-z_{0}\right) & N_{x\_start}\leq i<N_{x\_end}\\ x_{target} & i\geq N_{x\_end} \end{array}\right.$11:    where $\alpha \left(i\right)=\left\{\begin{array}{cc} 1 & \mathrm{if}\;∥y_{est}\left[i\right]∥<\epsilon_{yz}\;\mathrm{and}\;∥z_{est}\left[i\right]∥<\epsilon_{yz}\\ 0 & \mathrm{otherwise} \end{array}\right.$12:13:    **Lateral and Vertical Trajectory:**▹ Track target in y-axis and z-axis at $t=T_{hold}$14:    $y\left[i\right]\leftarrow \left\{\begin{array}{cc} y_{0} & i<N_{hold}\\ 0 & i\geq N_{hold} \end{array}\right.$ for i=0,…,N−115:    $z\left[i\right]\leftarrow \left\{\begin{array}{cc} z_{0} & i<N_{hold}\\ 0 & i\geq N_{hold} \end{array}\right.$ for i=0,…,N−116:    **return** {t,x,y,z}17: **end procedure**

To guarantee the stability of the resultant LQR controller, a positive definite input cost matrix R and state cost matrix Q are used to ensure the closed-loop system A−BK remains asymptotically stable, where K represents the stabilising gain. This condition can be summarised through Equation (25) [28]:(25)$$\begin{array}{c} \mathrm{rank}\left(\mathrm{obsv}\left(Q^{\frac{1}{2}},A\right)\right)=\min\left(m,n\right)\;,\;\mathrm{rank}\left(\mathrm{ctrb}\left(A,B\right)\right)=\min\left(m,n\right)\;, \end{array}$$
where m,n represent the dimensions of the resultant observability and controllability matrices. Given the state-space representation of the quadcopter in Equation (20) and the infinite-horizon cost function,(26)$$\begin{array}{c} J=\int_{0}^{\infty }\left(\bar{x}{\left(t\right)}^{T}Q\bar{x}\left(t\right)+u{\left(t\right)}^{T}Ru\left(t\right)\right)dt\;, \end{array}$$
the quadcopter plant is confirmed to be completely controllable and $\left(Q^{\frac{1}{2}},A\right)$ is confirmed to be completely observable. By then solving the continuous-time algebraic Riccati equation (CARE) to obtain a positive solution S, the optimal cost-to-go function $J^{*}={\bar{x}}^{⊤}S\bar{x}$ is obtained. The visual servoing controller is then implemented through the control law,(27)$$\begin{array}{cc} u & =-K\bar{x}\;, \end{array}$$
where the stabilising gain $K=R^{-1}\left(B^{T}S\right)$. The cost matrices are then determined through Bryson’s rule [29], with the maximum acceptable state errors and control inputs selected as follows:(28)$$\begin{array}{cc} \max(\bar{x}) & =\left[\begin{array}{cccccccccccc} 1.0 & 1.0 & 1.0 & 0.2 & 0.2 & 0.1 & 1.0 & 1.0 & 1.0 & 1.0 & 1.0 & 1.0 \end{array}\right]\\ \max(u) & =\left[\begin{array}{cccccc} 1.0 & 1.0 & 1.0 & 1.0 & 1.0 & 1.0 \end{array}\right]\;. \end{array}$$

The maximum allowable state errors are determined based on the physical constraints of the laboratory environment. Therefore, a maximum trajectory deviation of 1.0 m is established, complemented by more restrictive bounds on the quadcopter’s attitude to ensure stable flight characteristics. Although this configuration may yield suboptimal tracking performance without additional parameter tuning, the resulting control authority is deemed adequate for the approach manoeuvre under investigation. The constraints for the control input, u, are similarly established by imposing limits of 1.0 N and 1.0 N m on the translational and rotational control forces, respectively, during the approach phase.

### 3.2. Load Swing Attenuation Controller

A combined feedforward control–feedback control scheme is proposed to both track the optimised trajectory and provide local feedback. The feedforward control input $u_{0}(t)$ is generated through an inverse-dynamics approach from the optimised trajectory states, $x_{0}(t)$. A time-varying linear–quadratic regulator (TVLQR) is then designed to provide the feedback control input u(t). Given the nonlinear nature of quadcopter dynamics, the TVLQR allows linearisation of the quadcopter dynamics at each interpolated trajectory step. This yields a set of dynamic models that are valid for all states at all trajectory timesteps [26]. When combined with the inverse dynamics feedforward path, the control structure shown in Figure 4 is proposed. A set of one-dimensional Kalman filters is designed for velocity estimation from position and acceleration sources for each state. This design is chosen to permit both the individual tuning of noise characteristics for each axis and to reduce the computational cost from requiring 8×8 matrix operations to scalar operations.

#### 3.2.1. Feedforward Controller

The optimised trajectory obtained through minimising the constrained NLP problem in Equation (22) yields a set of reference states, $q_{\mathrm{ref}}$, ${\dot{q}}_{\mathrm{ref}}$, and ${\ddot{q}}_{\mathrm{ref}}$. Inverse dynamics is then used to compute the feedforward control input $u_{0}$ using the quadcopter–payload model:(29)$$\begin{array}{cc} u_{0} & =M\left(q_{\mathrm{ref}}\right){\ddot{q}}_{\mathrm{ref}}+C\left({\dot{q}}_{\mathrm{ref}},q_{\mathrm{ref}}\right){\dot{q}}_{\mathrm{ref}}+G\left(q_{\mathrm{ref}}\right)+D\left({\dot{q}}_{\mathrm{ref}}\right)\;. \end{array}$$

In the absence of parametric and structural uncertainty, as well as collocation error, the feedforward input force $u_{0}$ is sufficient to follow the optimised trajectory. However, given the non-zero collocation error and expected non-zero uncertainty in real systems, a tracking feedback controller is designed in the following section.

#### 3.2.2. Feedback Controller

The feedback controller serves to close the control loop and to minimise the effect of model uncertainties on the response of the feedforward controller. To track the optimised trajectory, TVLQR theory is leveraged to design the outer loop feedback controller. The forward dynamics of the quadcopter–payload system can be expressed as follows:(30)$$\begin{array}{ccc} \; & \dot{x} & =f(x,u),\\ \mathrm{where}, & x & =\left[\begin{array}{c} q\\ \dot{q} \end{array}\right],\;\dot{x}=\left[\begin{array}{c} \dot{q}\\ \ddot{q} \end{array}\right]. \end{array}$$
where x and u represent the state and control input vectors of the system. The optimised trajectory is then defined as $x_{0}\left(t\right),u_{0}\left(t\right)\forall t\in \left[t_{1},t_{2}\right)$. The tracking error between the optimised trajectory and current state is defined as $\bar{x}\left(t\right)=x\left(t\right)-x_{0}\left(t\right)$ and the input error between the feedforward input and feedback input as $\bar{u}\left(t\right)=u\left(t\right)-u_{0}\left(t\right)$. The linearised state-space representation of the error dynamics is then obtained via a first-order Taylor expansion:(31)$$\begin{array}{cc} \bar{\dot{x}}\left(t\right) & =f\left(x,u\right)-f\left(x_{0},u_{0}\right)\\ \; & \approx {\left.\frac{\partial f}{\partial x}\right|}_{\left(x_{0},u_{0}\right)}\bar{x}\left(t\right)+{\left.\frac{\partial f}{\partial u}\right|}_{\left(x_{0},u_{0}\right)}\bar{u}\left(t\right)\\ \; & =A\bar{x}\left(t\right)+B\bar{u}\left(t\right). \end{array}$$

To compute A(t) and B(t), the quadcopter equations of motion in Equation (9) are expressed in terms of $\dot{x}\left(t\right)$,(32)$$\begin{array}{c} \dot{x}=\left[\begin{array}{c} \dot{q}\\ M^{-1}\left(q\right)\left(Q-C\left(\dot{q},q\right)\dot{q}-G\left(q\right)-D\left(\dot{q}\right)\right) \end{array}\right]\;. \end{array}$$

Thus, the Jacobians are defined as follows:(33)$$\begin{array}{cc} A(t) & =\left[\begin{array}{cc} 0 & I\\ \frac{\partial M^{-1}}{\partial q}\left(N\right)+M^{-1}\left(-\frac{\partial G}{\partial q}-\frac{\partial D}{\partial q}-\frac{\partial C}{\partial q}\dot{q}\right) & M^{-1}\left(-\frac{\partial C}{\partial \dot{q}}\dot{q}-C\right) \end{array}\right], \end{array}$$(34)$$\begin{array}{cc} B(t) & =\left[\begin{array}{c} 0\\ M^{-1}B \end{array}\right], \end{array}$$
where $N=Q-C\left(\dot{q},q\right)\dot{q}-G\left(q\right)-D\left(\dot{q}\right)$. To determine the feedback gain at a timestep *k*, a discrete-time cost-to-go function is defined as follows:(35)$$\begin{array}{cc} J & =\Sigma_{k=0}^{N}\left({\bar{x}}_{k}^{T}Q{\bar{x}}_{k}+{\bar{u}}_{k}^{T}R{\bar{u}}_{k}\right)\;, \end{array}$$
where ${\bar{x}}_{k}=\bar{x}\left(t=kT_{s}\right)$, ${\bar{u}}_{k}=\bar{u}\left(t=kT_{s}\right)$, and $T_{s}$ represents the timestep between successive points on the optimised trajectory following interpolation. The optimal cost-to-go function $J^{*}$, which satisfies the Hamilton–Jacobi–Bellman equation, is determined as follows:(36)$$\begin{array}{c} J^{*}={\bar{x}}_{k}^{T}S_{k}{\bar{x}}_{k} \end{array}$$
where $S_{k}$ is the unique, positive-definite solution to the discrete-time differential Riccati equation calculated using $A_{k}=A\left(t=kT_{s}\right)$ and $B_{k}=B\left(t=kT_{s}\right)$:(37)$$\begin{array}{cc} S_{k-1} & =A_{k}^{T}S_{k}A_{k}-A_{k}^{T}S_{k}B_{k}{\left(R+B_{k}^{T}S_{k}B_{k}\right)}^{-1}B_{k}^{T}S_{k}A_{k}+Q\;,\\ \mathrm{where} & \;\;S_{N}=Q_{N}. \end{array}$$

Q≥0 and R>0 are defined as constant state and control cost matrices, respectively. These are kept unchanged at all timesteps to prioritise a uniform trade-off between tracking accuracy and control input across all timesteps. The control law is then introduced as follows:(38)$$\begin{array}{cc} u_{k} & =u_{0_{k}}-K{\bar{x}}_{k}\;,\\ \mathrm{where}\;\;K_{k} & =R^{-1}B_{k}^{T}S_{k}\;. \end{array}$$

In the following sections, the reachability, controllability, and observability of the proposed closed-loop system are examined.

### 3.3. Trajectory Verification

The feasibility of the optimised trajectory is verified by applying a finite-time feasibility analysis to compute a set of initial conditions that ensures the model stays within a specified goal state. This set is computed by leveraging methods found in [30] to first compute the stability of the goal state and then perform a simulation-based verification [31] to approximate the funnel where intermediate states both stay within state constraints and reach the goal state. For notation, this set is referred to as the invariant funnel.

#### 3.3.1. Goal State Stability

The approximated region of attraction is first derived around the goal state under TVLQR feedback. It can be demonstrated that there exists a Lyapunov function V(x) over a goal region $B(\rho_{G})$,(39)$$\begin{array}{cc} B(\rho_{G}) & :=\{0\leq V\left(x\right)\leq \rho_{G}\} \end{array}$$
where asymptotic stability is guaranteed if V(x) is positive definite, $\rho_{G}$ is a positive scalar, and $\dot{V}\left(x\right)<0$ in $B(\rho_{G})$. Following [30], the cost-to-go function is selected from Equation (36) at the goal sample k=G as the Lyapunov function, such that $V\left(\bar{x}\right)={\bar{x}}_{G}^{T}S_{G}{\bar{x}}_{G}$. To confirm that $\dot{V}\left(\bar{x}\right)<0$ holds for a given $\rho_{G}$, it is confirmed that a feasible $h({\bar{x}}_{G})$ can be found for the sum-of-squares (SOS) program [31] in Equation (40). A binary search for the largest $\rho_{G}$ is then performed such that $\dot{V}\left(\bar{x}\right)<0$ is maintained by the following:(40)$$\begin{array}{cc} \mathrm{find}\; & h({\bar{x}}_{G})\\ \mathrm{subject}\;\mathrm{to}\; & \dot{V}\left(\bar{x}\right)+h\left({\bar{x}}_{G}\right)\left(\rho_{G}-V\left(\bar{x}\right)\right)<\epsilon ,\\ \; & h({\bar{x}}_{k})\geq 0, \end{array}$$
where ϵ>0 and $h({\bar{x}}_{k})$ is a second-order polynomial. In addition, to leverage SOS, a cubic polynomial interpolation of our time-varying dynamics around the goal state $\left(x_{G}\;,u_{G}\right)$ is chosen.

#### 3.3.2. Funnel Simulation

To approximate the invariant funnel through simulation, a funnel parameter $\phi_{k}$ is found such that the quadcopter state at *k* lies within the hyper-ellipse bounded by the following:(41)$$\begin{array}{c} {\bar{x}}_{k}^{T}S_{k}{\bar{x}}_{k}<\phi_{k}\;,\;k=0\;\ldots \;N-1 \end{array}$$
where *N* represents the last timestep in the optimised trajectory. To initialise this simulation and provide sufficient coverage across the initial conditions bounded by Table 2, random initial states $x_{s}$ are considered within the initial bounds defined in Table 2. Then, forward time simulation is performed using the method in Section 4.1 under the control law in Equation (38). Upon completion of the simulation, if any of the state constraints in Table 2 are violated, or the final state does not lie in the goal region, the invariant funnel is shrunk based on the cost-to-go of the trajectory starting from k=0,(42)$$\begin{array}{c} \phi_{k,\;\mathrm{new}}=\min\left(\phi_{k},{\bar{x}}_{k}^{T}S_{k}{\bar{x}}_{k}\right)\;,\;k=0\;\ldots \;N-1 \end{array}$$

The approximated invariant funnel in Figure 5 is generated from 10,000 simulations with random initial states and takes $\rho_{G}=0.5746$ from the formal verification of the goal region of attraction. The pseudo-random initial states are generated on a uniform distribution with the mt19937 Mersenne Twister generator found in the C++ STL [32].

### 3.4. Controllability and Observability

The controllability of the linearised system is then investigated by computing the rank of the controllability Gramian. It is noted that the controllability Gramian satisfies the linear matrix differential equation,(43)$$\begin{array}{c} \dot{W_{c}}\left(t,t_{0}\right)=A\left(t\right)W_{c}\left(t,t_{0}\right)+W_{c}\left(t,t_{0}\right)A{\left(t\right)}^{T}+B\left(t\right)B{\left(t\right)}^{T},\;W_{c}\left(t_{0},t_{0}\right)=0, \end{array}$$
where $W_{c}(t,t_{0})$ is the controllability Gramian over the interval $\left[t_{0},t\right]$ [28]. The system is controllable over the interval $\left[t_{0},t_{f}\right]$ if the Gramian $W_{c}(t_{f},t_{0})$ is positive definite, or being full rank. It is then integrated through time, noting that the state transition matrix $W_{c}(t_{f},t_{0})$ limits to the zero matrix as it approaches the boundary time. The numerical integration is performed using a timestep of Δt=0.001 seconds over a finite interval $\left[t_{0},t_{f}\right]$ through the forward Euler integration,(44)$$\begin{array}{c} {\dot{W}}_{c,k+1}={\dot{W}}_{c,k}+\Delta t\left[A_{k}{\dot{W}}_{c,k}+{\dot{W}}_{c,k}A_{k}^{T}+B_{k}B_{k}^{T}\right], \end{array}$$
for k=0,1,…,N−1, where ${\dot{W}}_{c,0}=0$, ${\dot{W}}_{c,N}\approx W_{c}\left(t_{f},t_{0}\right)$, $t_{k}=t_{0}+k\Delta t$, Δt=0.001, $t_{0}=0$, $t_{f}=5$, and N=5000. The rank of the resultant controllability Gramian is then computed through singular value decomposition (SVD), such that(45)$$\begin{array}{c} \mathrm{rank}\left(W_{c,N}\right)=\mathrm{rank}\left(W\Sigma V^{T}\right)=\mathrm{rank}\left(\Sigma \right)\;. \end{array}$$

Likewise, the observability Gramian, *M*, is computed over the interval $\left[t_{0},t\right]$. In reference to [28], it is noted that the observability Gramian satisfies the linear matrix differential equation,(46)$$\begin{array}{c} \dot{M}\left(t,t_{1}\right)=-A\left(t\right)M\left(t,t_{0}\right)-M\left(t,t_{0}\right)A{\left(t\right)}^{T}-C\left(t\right)C{\left(t\right)}^{T},\;M\left(t_{1},t_{1}\right)=0\;, \end{array}$$
where C represents the output matrix $C=\left[\begin{array}{cc} I_{8\times 8} & 0_{8\times 8} \end{array}\right]$. A backward Euler integration is then performed with a timestep of Δt=0.001 seconds to obtain $M(t,t_{1})$. Similar to the controllability Gramian, SVD is used to determine the rank of the observability Gramian. The full rank $W_{c}$ and *M* indicate the controllability and observability of the LTV system across the entire trajectory with unbounded control actions.

## 4. Results

### 4.1. Model Simulation

The performance of the proposed LQR and inverse dynamic control strategies is first examined using the non-linear derived models of a quadcopter and a quadcopter–payload system, respectively. This allows for faster initial tuning of each controller without fully implementing the controllers on the off-board computer. A Runge–Kutta method iterative simulation is first performed on the derived models to evaluate the unforced response of the quadcopter–payload system. To permit this, the dynamics in Equation (9) are arranged as a differential-algebraic equation of index one.(47)$$\begin{array}{cc} \; & x=\left[\begin{array}{c} q\\ \dot{q} \end{array}\right],\;\dot{x}=\left[\begin{array}{c} \dot{q}\\ \ddot{q} \end{array}\right] \end{array}$$
where $q\in R^{8\times 1}$, $\dot{q}\in R^{8\times 1}$, $x\in R^{16\times 1}$, and $\dot{x}\in R^{16\times 1}$. This allows the state dynamics to be arranged in the following form:(48)$$\begin{array}{cc} E\left(x\right)\dot{x} & =f(x) \end{array}$$
where(49)$$\begin{array}{cc} E(x) & =\left[\begin{array}{cc} I & 0\\ 0 & M(q) \end{array}\right],\;f\left(x\right)=\left[\begin{array}{c} \dot{q}\\ -C\left(q,\dot{q}\right)\dot{q}-G\left(q\right)-D\left(q,\dot{q}\right)\;. \end{array}\right] \end{array}$$

The system dynamics are then numerically integrated. For the following simulations and flight tests, Table 4 lists the parameters used (referenced from an Alien 450 quadcopter).

### 4.2. Approach Task

This section discusses the performance of the approach task, which is characterised through two testing methodologies—a simulation of the LQR controller tracking an AprilTag target imaged by a virtual camera stream within the Gazebo simulation environment, and a physical flight test conducted within the autonomous systems laboratory. For all tests, the reference trajectory is generated through Algorithm 1 along with the AprilTag CUDA v1.0 implementation detailed in Section 3.1. The camera intrinsics are obtained through a checkerboard pattern calibration.

#### 4.2.1. Gazebo Simulation Environment

To minimise the risk of hardware damage during in-flight testing, the dimensions of the UNSW Autonomous Systems Laboratory are replicated in a Gazebo environment shown in Figure 6. A single AprilTag is then placed at the expected real-world coordinates, and a modelled quadcopter is inserted at the expected take-off point. A detailed comparison between the CAD model used in Gazebo and the actual quadcopter hardware is provided in Section B.2. To minimise software variation between simulation and real-world environments, the PX4 Autopilot framework [33] is utilised in both simulation and real-world environments. Virtual and physical sensors are selected when switching between the environments.

#### 4.2.2. Real-World Flight Test Configuration

The physical validation is performed using a quadcopter based on the Alien 450 (HJ, Shenzhen, China) platform equipped with a RealSense D400 camera (Intel Corporation, Santa Clara, CA, USA) operating without depth information. The camera is rigidly mounted to the quadcopter frame above the propellers to maintain target visibility during approach manoeuvres. The test environment consists of a 5 m×5 m×2.4 m flight volume within the Autonomous Systems Laboratory, as illustrated in Figure 7. A single 36h11 AprilTag with a width of 0.25 m is mounted on a rigid stand at a height of 1.5 m. As AprilTags allows six-degree-of-freedom pose estimation without stereo camera requirements through known physical tag dimensions and camera intrinsics, it only requires a single AprilTag to determine the quadcopter pose relative to the tag.

### 4.3. Approach Test Results

This section verifies the regulating performance of the LQR controller used for trajectory tracking. An initial position error of(50)$$x_{\mathrm{error}(0)}=1.82\;m,\;y_{\mathrm{error}(0)}=-0.065\;m,\;z_{\mathrm{error}(0)}=-0.33\;m$$
is considered to test both the Gazebo simulation and the real-world test. This initial position error is obtained through the relative distance between the take-off point in the middle of the laboratory and the placement of the AprilTag.

The validation results shown in Figure 8 demonstrate strong agreement between the simulation and real-world implementation across all three axes. For the *y*-axis, both the simulation and real-world responses maintain reference tracking with minimal deviation, exhibiting peak errors below 0.05 m throughout the test duration, with well-bounded oscillations indicating stable lateral control. For the *z*-axis, while both the simulation and real-world tests successfully converge from their initial offsets to the reference trajectory within 10 s, the real-world system demonstrates notably faster initial response and reaches steady-state earlier than the simulation. This is largely due to parametric and structured uncertainties between the CAD model and the real-world quadcopter. The *x*-axis yields successful tracking performance during the approach task, with both simulation and real-world systems closely following the reference trajectory from 1.82 m to 0.60 m, achieving a steady-state response at the 30 s mark.

### 4.4. Load Swing Attenuation

The stabilisation algorithm aims to satisfy the problem statement of ensuring flight stability after the release of the extension arm. Furthermore, the control objective necessitates attenuating the load-swing motion faster than the natural damping when no controller is applied.

#### Model Simulation

Due to the low velocity of the optimised trajectory, the linear drag that affects the quadcopter’s translation is zeroed for these simulations. The response of the TVLQR controller is first examined, which provides local feedback to the optimised trajectory with nominal parameters.

As expected, by tracking the optimised trajectory in Figure 9, the proposed controller successfully attenuates the load-swing motion of the pendulum. As the simulated behaviour does not introduce uncertainties, the optimal acceleration input is expected to track the trajectory perfectly. However, minor tracking errors are expected due to the effects of the collocation error between collocation points, where the optimised trajectory does not entirely satisfy the system dynamics between the collocation points. Furthermore, it is expected that numerical errors introduce discrepancies between the optimiser dynamics and the RK4 simulation. This indicates that using an inverse dynamics-based control strategy with a TVLQR feedback controller can successfully track the optimal reference trajectory for load-swing attenuation. Compared to the unforced response of the payload swing, the load-swing motion is successfully forced to zero before the simulation ended. Some model uncertainties are then introduced to the simulation. By increasing the quadcopter and load masses by a factor of 0.8 and 1.2, it is shown in Figure 10 that the feedback controller is still able to successfully complete the attenuation task.

Despite the large parametric uncertainty, it is demonstrated that the TVLQR feedback is able to successfully attenuate the load swing. However, it is noted that large parameter uncertainties can cause the feedback controller to be incapable of fully tracking the *z*-position state during the trajectory. This is due to the large variation in mass of the system (≈0.4 kg), leading to a large residual force between the trajectory-determined force and required force in the *z* direction. Finally, a wind disturbance is modelled as a Gaussian distributed force with a mean and standard deviation of $\bar{x}=5N\;,\sigma =1.0$. It can be seen from Figure 11 that the load-swing objective is successfully executed, while abiding by the optimised trajectory.

## 5. Discussion

This study has demonstrated the effectiveness of the proposed AprilTag-based pose estimation and time-invariant LQR feedback for achieving visual servoing in approaching the payload. Furthermore, it has been shown that trajectory optimisation, combined with TVLQR local feedback, permits load-swing attenuation in quadcopter–payload systems under a variety of disturbances and model uncertainties.

The effectiveness of the visual servoing controller has been confirmed through numerical simulations, Gazebo-based software-in-the-loop simulations, and real-world experiments. The quadcopter–payload system has been successfully stabilised following the release of the payload arm via the proposed feedforward and feedback control scheme, outperforming natural damping. The simulations with nominal parameters have shown near-perfect trajectory tracking, with minor errors attributed to collocation and numerical discrepancies. The TVLQR controller combined with an optimised trajectory has been able to attenuate the load swing under significant parametric uncertainty (e.g., a 20% increase in quadcopter and load masses). These results have validated the robustness of the proposed control framework and its applicability to real-world scenarios involving model uncertainties.

## Figures and Tables

**Figure 1 sensors-25-05518-f001:**
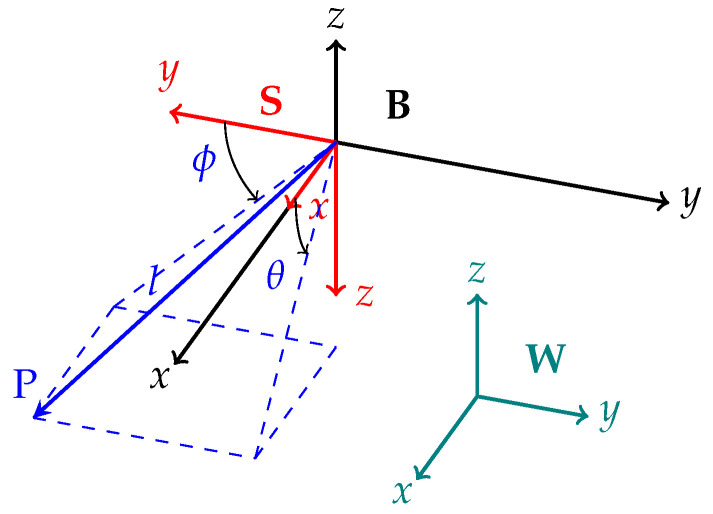
Reference frames.

**Figure 2 sensors-25-05518-f002:**
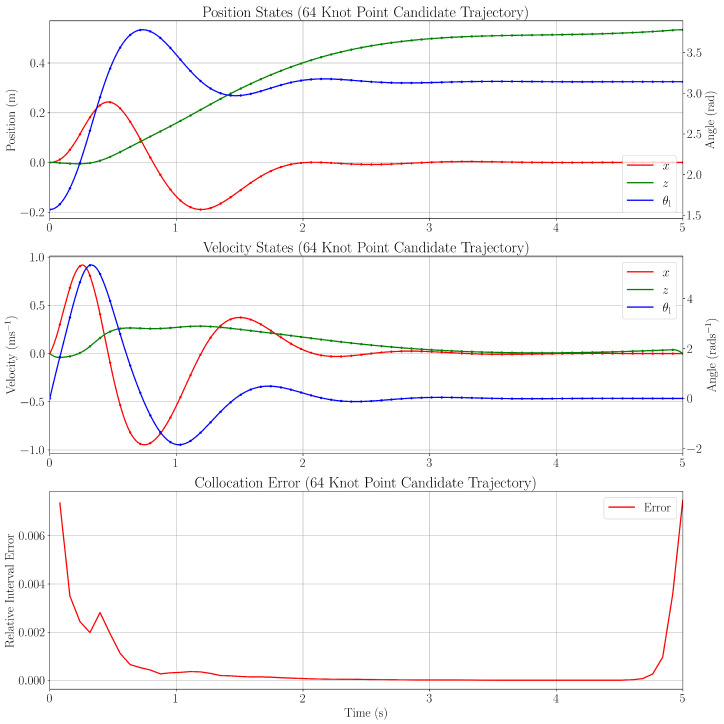
Optimised trajectory.

**Figure 3 sensors-25-05518-f003:**
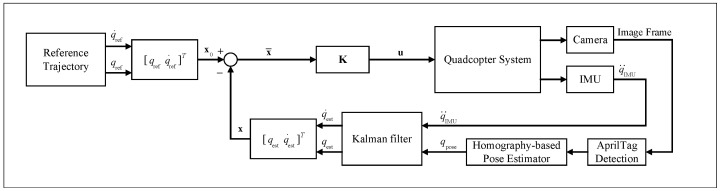
Pose-based visual servoing control block diagram.

**Figure 4 sensors-25-05518-f004:**
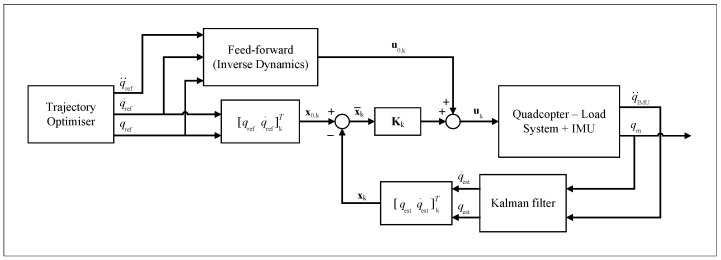
Load swing attenuation control block diagram.

**Figure 5 sensors-25-05518-f005:**
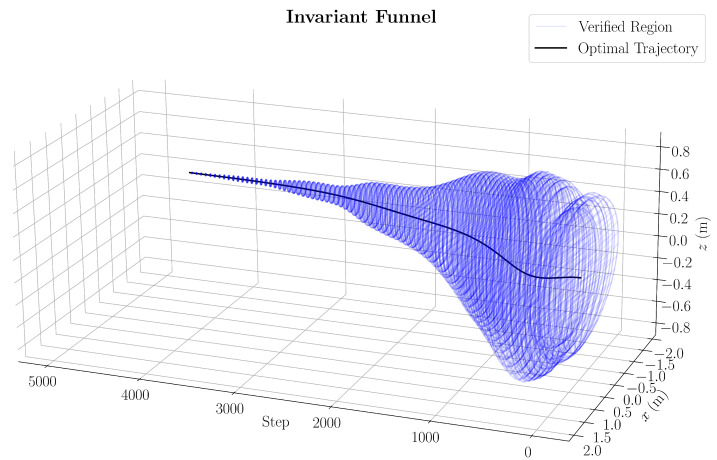
Invariant funnel of the optimised trajectory under TVLQR feedback.

**Figure 6 sensors-25-05518-f006:**
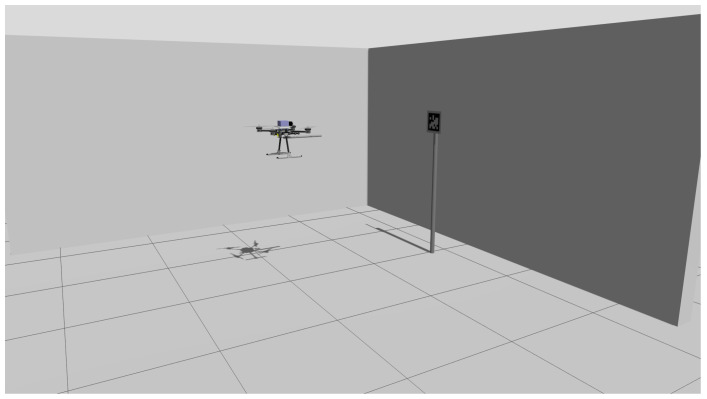
Simulated environment layout showing the virtual quadcopter initial position and AprilTag target location.

**Figure 7 sensors-25-05518-f007:**
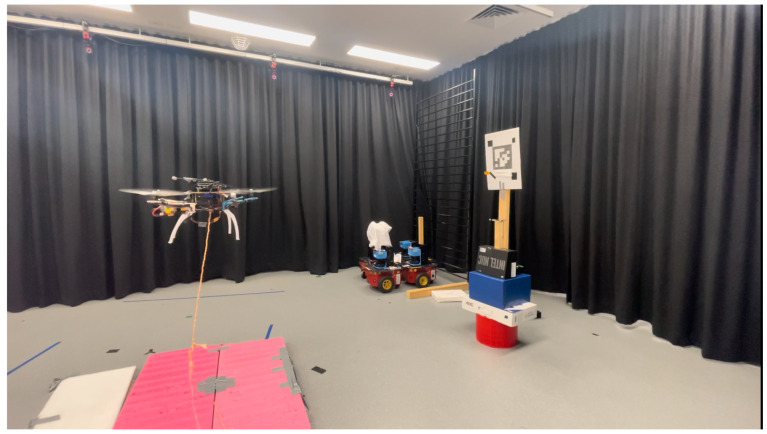
Real-world environment layout showing the quadcopter’s initial position and AprilTag target location.

**Figure 8 sensors-25-05518-f008:**
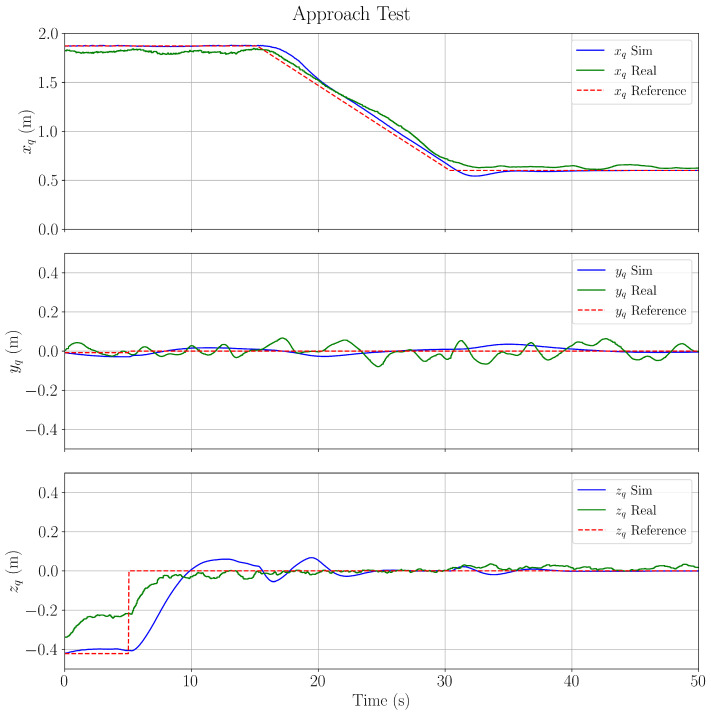
Validation results for the simulated versus real-world visual servoing-based trajectory following.

**Figure 9 sensors-25-05518-f009:**
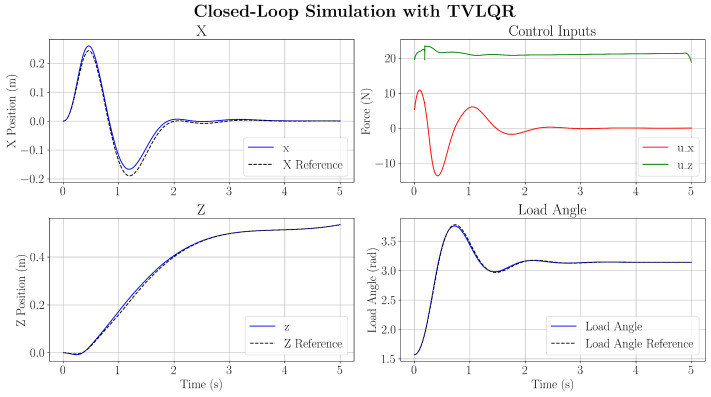
Closed-loop controller response.

**Figure 10 sensors-25-05518-f010:**
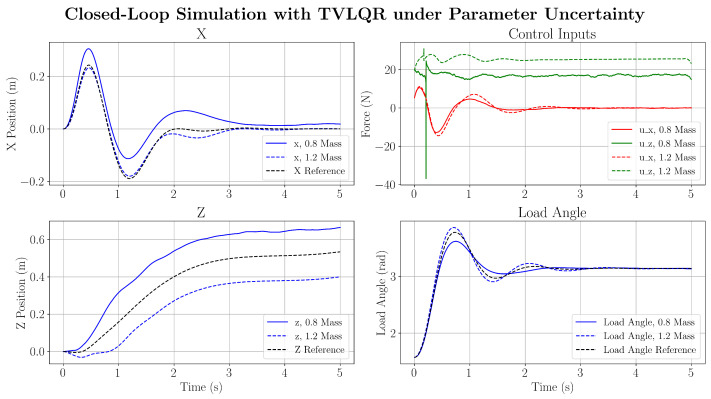
Closed loop controller response under parametric uncertainty.

**Figure 11 sensors-25-05518-f011:**
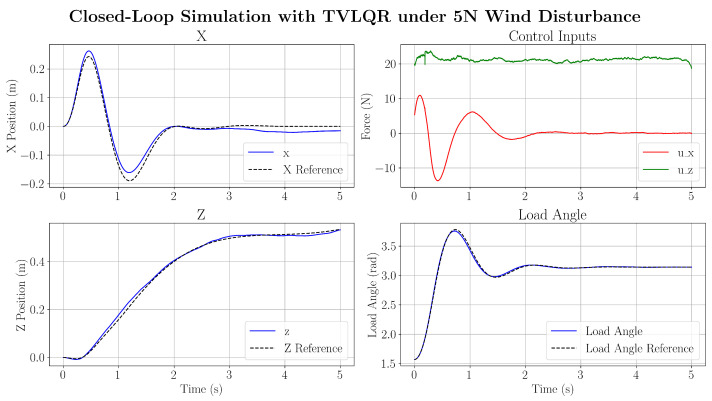
Closed-loop controller response under wind disturbance force.

**Table 1 sensors-25-05518-t001:** Trajectory generation input constraints.

Input	u (Min)	u (Max)
$F_{x}\;\left(N\right)$	−30.0	30.0
$F_{y}\;\left(N\right)$	−30.0	30.0
$F_{z}\;\left(N\right)$	0.0	25.0
$\tau_{\phi }\;\left(N\;m\right)$	−1.0	1.0
$\tau_{\theta }\;\left(N\;m\right)$	−1.0	1.0
$\tau_{\psi }\;\left(N\;m\right)$	−1.0	1.0
$\tau_{\theta l}\;\left(N\;m\right)$	0.0	0.0
$\tau_{\psi l}\;\left(N\;m\right)$	0.0	0.0

**Table 2 sensors-25-05518-t002:** Trajectory generation state constraints.

State	x_i_ (Min)	x_i_ (Max)	x_n_ (Min)	x_n_ (Max)	x_f_ (Min)	x_f_ (Max)
$x_{q}\;\left(m\right)$	0.0	0.0	−2.0	2.0	−2.0	2.0
$y_{q}\;\left(m\right)$	0.0	0.0	0.0	0.0	0.0	0.0
$z_{q}\;\left(m\right)$	0.0	0.0	−1.0	2.0	−1.0	2.0
$\phi_{q}\;\left(\mathrm{rad}\right)$	0.0	0.0	−0.5·π	0.5·π	0.0	0.0
$\theta_{q}\;\left(\mathrm{rad}\right)$	0.0	0.0	−0.5·π	0.5·π	0.0	0.0
$\psi_{q}\;\left(\mathrm{rad}\right)$	0.0	0.0	−0.5·π	0.5·π	0.0	0.0
$\phi_{l}\;\left(\mathrm{rad}\right)$	0.0	0.0	−π	π	0.0	0.0
$\theta_{l}\;\left(\mathrm{rad}\right)$	0.5·π	0.5·π	0.5·π	1.5·π	π	π
${\dot{x}}_{q}\;\left(m\;s^{-1}\right)$	0.0	0.0	−5.0	5.0	0.0	0.0
${\dot{y}}_{q}\;\left(m\;s^{-1}\right)$	0.0	0.0	−2.0	2.0	0.0	0.0
${\dot{z}}_{q}\;\left(m\;s^{-1}\right)$	0.0	0.0	−5.0	5.0	0.0	0.0
${\dot{\phi }}_{q}\;\left(\mathrm{rad}\;s^{-1}\right)$	0.0	0.0	−10.0	10.0	0.0	0.0
${\dot{\theta }}_{q}\;\left(\mathrm{rad}\;s^{-1}\right)$	0.0	0.0	−10.0	10.0	0.0	0.0
${\dot{\psi }}_{q}\;\left(\mathrm{rad}\;s^{-1}\right)$	0.0	0.0	−10.0	10.0	0.0	0.0
${\dot{\phi }}_{l}\;\left(\mathrm{rad}\;s^{-1}\right)$	0.0	0.0	−∞	∞	0.0	0.0
${\dot{\theta }}_{l}\;\left(\mathrm{rad}\;s^{-1}\right)$	0.0	0.0	−∞	∞	0.0	0.0

Min and Max values for each state at the initial ($x_{i}$), knot ($x_{n}$), and final ($x_{f}$) time points.

**Table 3 sensors-25-05518-t003:** AprilTag CUDA v1.0 vs AprilTag 3 v3.4.4 performance comparison.

Library	System	Tag36h11 640 × 480	Tag36h11 1280 × 720	Tag36h11 1600 × 900
AprilTag CUDA v1.0	System: NVIDIA Jetson Orin Nano 8 GB CPU: 6-core ARM A78AE @ 1.98 GHz GPU: 1024-core Ampere	μ: 2.242 ms σ: 0.331 ms	μ: 13.113 ms σ: 0.383 ms	μ: 17.295 ms σ: 0.485 ms
AprilTag3 v3.4.4	System: NVIDIA Jetson Orin Nano 8 GB CPU: 6-core ARM A78AE @ 1.98 GHz GPU: Unused	μ: 5.398 ms σ: 1.006 ms	μ: 46.839 ms σ: 7.346 ms	μ: 55.491 ms σ: 9.585 ms
AprilTag3 v3.4.4	System: Intel NUC5i7RYB CPU: 2-core Intel Core i7-5557U @ 3.10 GHz GPU: Unused	μ: 2.833 ms σ: 0.627 ms	μ: 42.172 ms σ: 2.377 ms	μ: 45.793 ms σ: 1.953 ms

**Table 4 sensors-25-05518-t004:** Physical parameters.

Symbol	Definition	Value	Unit
*g*	Gravitational acceleration	9.810	m s ^−2^
$m_{l}$	Load mass	0.100	kg
$m_{q}$	Quadrotor mass	2.064	kg
*l*	Arm Length	0.500	m
$k_{qd_{x}}$, $k_{qd_{y}}$, $k_{qd_{z}}$	Quadcopter Damping Coefficients	0.000	N m^−1^
$k_{qd_{l}}$	Payload Damping Coefficient	0.0129	N m rad^−1^
$i_{xx}$	Moment of inertia about x-axis	0.00291	kg m^2^
$i_{yy}$	Moment of inertia about y-axis	0.00291	kg m^2^
$i_{zz}$	Moment of inertia about z-axis	0.00552	kg m^2^

## Data Availability

The raw data supporting the conclusions of this article will be made available by the authors upon request.

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
