# Peer review of "Load-Swing Attenuation in a Quadcopter–Payload System Through Trajectory Optimisation"

_sensors, 2025, doi:10.3390/s25175518_

Round 1
Reviewer 1 Report
Comments and Suggestions for Authors
In the paper, the performance of the visual servoing approach was verified by both numerical simulations and a physical quadcopter implementation, which leverages a CUDA-accelerated tag detection algorithm for real-time pose estimation of the target. And the performance of the load-swing tenuation was also verified by numerical simulations. These studies have certain promotional significance for the advancement of multi-rotor quadcopter technology and sensing capabilities.
In multi-rotor quadcopter technology, trajectory optimization is not an original innovation, but it plays an important guiding and promoting role in attenuating load swing in the payload system.The research in this paper provides the possibility of breaking through the constraints on the application of quadcopters in long-distance delivery scenarios. The conclusions of the paper are consistent with the evidence and arguments presented.
- The formatting of the scientific paper is not rigorous enough. The text alternates between third-person and first-person narration, which does not conform to the writing requirements of scientific papers.
- The presentation of Figure 2 is very poor.
- For such an interesting study, it would be better to add experimental verification to improve the entire research.
Author Response
Comment 1: The formatting of the scientific paper is not rigorous enough. The text alternates between third person and first-person narration, which does not conform to the writing requirements of scientific papers.
Response 1: Agreed. We have fully revised the English and modified the text across the paper to provide consistency in the use of proper scientific language.
Comment 2: The presentation of Figure 2 is very poor.
Response 2: Agreed.
- We have ensured that the trajectory graphs now start and end on the vertical borders.
- We have ensured that the legend now does not include duplicate entries.
- We have increased the thickness of the graph lines and scatter plot markers.
- The modified and improved Figure 2 is provided on Page 8
Comment 3: For such an interesting study, it would be better to add experimental verification to improve the entire research.
Response 3:
We're pleased to confirm alignment on this point, and we're excited to share that experimental verification of the swing attenuation is already in our pipeline for upcoming research. To maintain a clear and focused narrative in the current paper, we've chosen to present these experimental results in a dedicated future publication where they can receive the thorough treatment they deserve.
Reviewer 2 Report
Comments and Suggestions for Authors
Paper Summary
This paper presents a control strategy for mitigating load swing in quadcopter-payload systems during and after payload retrieval. The authors first implement a visual servoing approach using a CUDA-accelerated AprilTag detection algorithm and a time-invariant LQR (TILQR) controller to enable precise positioning during payload acquisition. Following the acquisition, the paper proposes a trajectory optimization framework using direct collocation to compute a swing-attenuating trajectory for the quadcopter.
A time-varying linear quadratic regulator (TVLQR) with inverse-dynamics feedforward control is then designed to track the optimized trajectory. The authors formulate and verify the controllability, observability, and feasibility of the closed-loop system. Performance is evaluated through simulations that incorporate parametric uncertainty and wind disturbances, and the approach is validated through real-world experiments using an Alien 450 quadcopter. The study includes both model-based simulation and real-world validation of the visual servoing and load-swing attenuation mechanisms.
Detailed Comments
- L87-L94. In Section 1.1-1.2, though related work is reviewed thoroughly; the authors may consider emphasizing how their proposed solution improves upon or differentiates from these prior approaches in terms of robustness, control performance, or applicability to large swing angles.
- L109: Could you clarify the modeling assumptions regarding the payload? More specifically, is the payload treated as an idealized point mass that does not influence the quadcopter beyond its swing dynamics?
- Section2.4: If so, could you elaborate on the sensitivity of the trajectory optimization algorithm to the choice of initial conditions?
- L178, L183: Could you provide more details on the CUDA-based AprilTag marker detection implementation?How does the performance compare to the standard CPU-based version during real-time operation?
- L206: Could you elaborate more on the selection of the maximum acceptable state and control input values?
- L216: The proposed control algorithm uses a set of 1D Kalman filters for velocity estimation. could you clarify the reasoning behind this choice (instead of using multi-dimensional kalman filter)?
- Chapter4: This paper would benefit from a clearer emphasis on the role of GPU acceleration. Specifically, please consider providing performance metrics such as frame rate, detection latency, or speedup compared to the CPU-only AprilTag implementation.
- Reference: Please double check references to follow the reference guideline https://www.mdpi.com/authors/references, e.g. the format of L485, L492 are inconsistent with L443, L451.
Author Response
Please see the attachment. A PDF file has been uploaded.
However, the replies are also provided here just in case:
Reviewer 2
Comment 1: L87-L94. In Section 1.1-1.2, though related work is reviewed thoroughly; the authors may consider emphasizing how their proposed solution improves upon or differentiates from these prior approaches in terms of robustness, control performance, or applicability to large swing angles.
Response 1: Agree. We have made the following additions to our contributions section from L88-L100 on page 3.
“The state-of-the-art is extended by presenting a feasible control solution for quadcopter flight while carrying a swung payload from an initially horizontal position. This extensively differs from existing literature by presenting a feasible and optimal solution to generate the shortest path from a worst-case initial swing-angle to the downward equilibrium condition. In existing literature, the techniques that employ a single linearisation in the downward payload position no longer produce accurate approximations when the payload begins in a horizontal position. In addition, they do not offer solutions with notions of optimality in terms of the shortest attenuation path. In this article, an application of pose-based visual servoing is then presented using time-invariant LQR (TILQR) feedback and a custom CUDA [22] port of the AprilTag detection library [23] to enable the quadcopter to approach the payload. The CUDA implementation replaces the thresholding, segmentation and clustering steps to yield an image-to-pose latency improvement by more than three times compared to public CPU implementations on similar grade hardware. ”
Comment 2: L109: Could you clarify the modeling assumptions regarding the payload? More specifically, is the payload treated as an idealized point mass that does not influence the quadcopter beyond its swing dynamics?
Response 2: Agree. We had originally included the modelling assumptions regarding the payload in the appendix, but we have now moved the section up to the modelling assumptions section (Section 2.1).
Please see the following edits in L120-L126 on page3:
“The effect of the payload on the quadcopter's dynamics is modelled under the assumption that the load behaves as a point mass on the end of a spherical pendulum. The spherical pendulum is then assumed to be mechanically connected to the center of mass of the quadcopter structure. In addition, the mass of the pendulum arm is considered negligible while the pendulum arm is infinitely stiff and the suspension point is assumed frictionless, allowing pendulum arm motion to behave as an ideal spherical pendulum.”
Comment 3: Section2.4: If so, could you elaborate on the sensitivity of the trajectory optimization algorithm to the choice of initial conditions?
Response 3: We'd like to clarify an important distinction in Section 2.4. When we discuss convergence to similar trajectories, we're referring to different problem setups rather than initial conditions. Specifically, our statement - " While, it is also known that this technique suffers from returning sub-optimal solutions due to convergence to local minima, we were able to converge to the same solution across a range of problem initializations." - addresses the state and control limits we establish for the optimization problem, not the starting positions themselves.
This distinction matters because our quadcopter's position is always measured relative to the tag, which is considered as (0, 0, 0). This reference frame choice eliminates concerns about sensitivity to different starting locations. Of course, real-world disturbances during flight can still push the system off its planned path. We thoroughly address this practical concern through our region of attraction analysis in Section 3.3, which demonstrates the system's robustness to such deviations.
To address this concern, we made the following edits to the paper in L157-L158 on page 5:
“While it is also known that this technique suffers from returning sub-optimal solutions due to convergence to local minima, convergence to the same solution across a range of problem initialisations can still be obtained. Furthermore, the reference frame for the trajectory is always considered to be referenced to the parcel itself. Therefore, the initial pose of the system is always taken as (0, 0, 0)m.”
Comment 4: L178, L183: Could you provide more details on the CUDA-based AprilTag marker detection implementation? How does the performance compare to the standard CPU-based version during real-time operation?
Response 4: Great point. We appreciate your interest in the GPU acceleration of the AprilTag3 library. We originally opted to explore this further in future works, but we have included a summary of the detection pipeline and performance metrics on flight hardware in Section 3.1.1 “AprilTag Detection Performance”.
Please see the edits below in L212-L228 on page 9,
“Table 3 provides a comparison between the performance of the CPU-based AprilTag3 v3.4.4 versus AprilTag CUDA. The CPU-based AprilTag3 v3.4.4 is profiled on both the Intel NUC and Jetson Orin Nano platforms using four threads with no decimation and a maximum Hamming bit error of two. AprilTag CUDA is profiled with identical decimation and error parameters. For the CPU-based quadrilateral decoding and pose estimation stages following CUDA parsing, single-threaded operation is used to optimise performance as the low quadrilateral count makes multi-threading unnecessary. The executable is run with 10000 iterations at each resolution to determine the execution time average (μ) and standard deviation (σ) over detections.
It can be seen from the results in Table 3 that at low resolutions (640x480) there is a small improvement in using AprilTag CUDA compared to AprilTag3 v3.4.4. However, at the desired (1280x720) resolution, an astounding 322% improvement in the GPU implementation via AprilTag CUDA can be observed. These results are similarly observed at the 1600x900 resolution (264% improvement). Furthermore, off-loading the detection task to the GPU minimises OS-scheduler effects as seen by the large standard deviations values when executing on the CPU.”
Comment 5: L206: Could you elaborate more on the selection of the maximum acceptable state and control input values?
Response 5: Agreed that this section required more elaboration. Please see the additions below to Section 3.1.2 in L256-L263 on page 11:
“The maximum allowable state errors are determined based on the physical constraints of the laboratory environment. Therefore, a maximum trajectory deviation of 1.0 m is established, complemented by more restrictive bounds on quadcopter attitude to ensure stable flight characteristics. Although this configuration may yield suboptimal tracking performance without additional parameter tuning, the resulting control authority is deemed adequate for the approach manoeuvre under investigation. The constraints for the control input, u are similarly established by imposing limits of 1.0 N and 1.0 N m on the translational and rotational control forces, respectively, during the approach phase. ”
Comment 6: L216: The proposed control algorithm uses a set of 1D Kalman filters for velocity estimation. could you clarify the reasoning behind this choice (instead of using multi-dimensional Kalman filter)?
Response 6: We initially chose a set of 1D Kalman filters out of implementation simplicity and reduction of computational requirements from needing to perform 8x8 matrix operations to simply requiring parallelizable scalar math. Furthermore, having a set of 1D filters allows all movement axes to be tuned individually, rather than having to consider cross-axis effects.
We have updated Section 3.2 to elaborate this in L274-L276 on page 11.
“A set of 1-dimensional Kalman filters are designed for velocity estimation from position and acceleration sources for each state. This design is chosen to permit both the individual tuning of noise characteristics for each axis, and to reduce computational cost from requiring 8 × 8 matrix operations to scalar operations.”
Comment 7: Chapter4: This paper would benefit from a clearer emphasis on the role of GPU acceleration. Specifically, please consider providing performance metrics such as frame rate, detection latency, or speedup compared to the CPU-only AprilTag implementation.
Response 7: We appreciate and agree with the comment. We have added more emphasis on the implementation and results of GPU acceleration in Response 4.
Comment 8: Reference: Please double check references to follow the reference guideline https://www.mdpi.com/authors/references, e.g. the format of L485, L492 are inconsistent with L443, L451.
Response 8: Agreed. We have reviewed the bibliography and ensured that it now meets MDPI Sensors formatting requirements. Main edits include:
- Ensuring the correct BibTex tags are attached to entries.
- Ensuring author names are formatted in Last, F.M. notation.

Reviewer 3 Report
Comments and Suggestions for Authors
See you attachment

Author Response
Please find the attachment. A PDF file is uploaded.
However, the reposes are also provided here just in case:
Reviewer 3
Comment 1: The authors have given the mass of the quadcopter and some of its parameters and also presented a mathematical model and Gazebo requires a CAD model to run the simulations. Was a CAD model used? If so it should have been presented, at least in the form of a photo. It is interesting to note how the authors added a "grabber" with which it was possible to study the interference affecting the drone in the Gazebo environment.
Response 1:
Thank you for your valuable feedback and for highlighting this important aspect of our work. We appreciate your interest in the CAD model used for our Gazebo simulations.
We are pleased to confirm that a CAD model was indeed developed for this study, and we agree that its visualization would enhance the clarity of our methodology. In response to your suggestion, we have now included an image of the CAD model alongside a photograph of the real-life quadcopter used in our experiments in both Figure 6, 7 on page 16 in the revised section 4.2.2, and Appendix C n page 23. These additions will help readers better understand the physical system and its digital representation in the simulation environment.
Regarding the grabber mechanism you noted, we would like to clarify that the extension arm mentioned in the manuscript was modeled as a rod on a cylindrical joint. This modeling approach allowed us to effectively study the interference effects on the drone dynamics within the Gazebo environment while maintaining computational efficiency and simulation accuracy. To this end, the grabber itself was not modelled as we did not study the “grabbing” of the AprilTag itself.
We believe these additions and clarifications strengthen the manuscript by providing readers with a more complete picture of both the experimental setup and simulation framework. Thank you once again for this constructive comment, which has helped us improve the presentation of our work.
Comment 2: In Figure 6, the authors perform evaluations between the real environment, the Gazebo environment and the arbitrary data. There is a complete lack of information on how this was carried out. Because if there is a real-world environment (real-world test), how was this realized? Was a drone flight performed. If so, what kind of drone and with what camera? Has the weight of the camera been taken into account? Is the drone consistent with what it was in the Gazebo? Where was the camera mounted and how was the environment chosen (using AprilTag requires the tags to be spread across the space of the drone). I can't find information on what the environment looked like for this experiment, how many tags there were and how distributed. Was the environment model in the Gazebo a representation of this in reality?
Response 2: Thank you for your valuable feedback on evaluations between the simulation and real environments. We split your questions and provided a reply to each indivifually as blow
- Because if there is a real-world environment (real-world test), how was this realized?
- Was a drone flight performed. If so, what kind of drone and with what camera?
- Has the weight of the camera been taken into account?
- Is the drone consistent with what it was in the Gazebo?
- Where was the camera mounted and how was the environment chosen?
- (using AprilTag requires the tags to be spread across the space of the drone)? I can't find information on what the environment looked like for this experiment, how many tags there were and how distributed?
- Was the environment model in the Gazebo a representation of this in reality?
Responses to each sub-question:
- We conducted real-world testing in the Autonomous Systems Laboratory at UNSW. A detailed description of this environment and the experimental execution has been added to Section 4.2.2 on page 16.
- Flight tests were performed using an Alien 450 quadcopter platform equipped with an Intel RealSense D400 camera operating in monocular mode (depth information was not utilized). This information has been incorporated into Section 4.2.2, with additional details regarding the quadcopter configuration provided in Appendix C.
- The camera weight was incorporated into our analysis and included in the total quadcopter mass. A detailed examination of this consideration is presented in Appendix C.
- While there are some differences between the Gazebo implementation and the physical construction of the drone, these have been documented in Appendix C. Importantly, we maintained consistency in core structural elements including frame geometry, rotor placement, and horizontal arm configuration, between our simulation model and physical prototype. This approach ensured the validity of our dynamic analysis while optimizing computational performance in the Gazebo environment. The convergence between our simulation and real-world test results, as demonstrated in Figure 8, further validates this methodology.
- The camera was mounted above the propellers at the front of the quadcopter, as illustrated in Figure A1 in the Appendix C on page 23 and described in Section 4.2.2 . The indoor flying space within the Autonomous Systems Laboratory at UNSW was selected as it provided a controlled and safe testing environment.
- We appreciate your observation regarding AprilTag pose estimation. As clarified in our revised text, our application successfully utilized a single AprilTag for pose estimation, given the known physical dimensions and camera intrinsic parameters. Following detection of the tag's image coordinates, pose estimation was achieved using established homography-based or PnP algorithms, as discussed in Section 4.2.2. The image coordinate detection process is detailed in Appendix B on pages 22-23, which also presents our GPU acceleration implementation of the AprilTag detection library. Additionally, we have added a new subsection (Section 3.1.1) on page 8-9 comparing AprilTag detection performance across platforms, profiling CPU-based AprilTag3 v3.4.4 on both Intel NUC and Jetson Orin Nano platforms, alongside AprilTag CUDA on the Jetson Nano platform using identical detection parameters.
- The environment model accurately reflects the dimensions of the real-world testing space. The relative positioning between the quadcopter and target was maintained consistently across both simulation and laboratory environments. Visual comparisons demonstrating this correspondence are provided in Figures 6 and 7 on page 16.
Comment 3: It has not been explained what a "Trajectory" is (fig 6)? Is it a preset trajectory or is it an arbitrarily measured trajectory (if so how was it measured). If not, what IMU sensor was placed on the drone to consider that the data from the drone's flight is data to which reference can be made? IMU sensors are subject to integration error and it may be that the data collected from the overflight is at variance with the true values of the drone's position.
Response 3:
Thank you for raising this point. We have fully revised Section 3.1.2 to provide greater clarity regarding the trajectory generation process with an added Algorithm. To clarify, the trajectory was dynamically generated in real-time from the drone's current pose to a target position directly in front of the AprilTag.
We would like to emphasize that the IMU was not used for position estimation through second-order integration at any point in our approach. Instead, the generated trajectory was tracked using a combination of localization data: position estimates were obtained from the AprilTag pose estimation, while the IMU contributed exclusively to velocity estimation. Specifically, the IMU-derived acceleration was processed through the 1D Kalman filter mentioned in our paper, which fused the IMU-based acceleration measurements with vision-based position data to produce velocity estimates.
Please find below the edits in L230-L243 on pages 9-10
“To maintain the AprilTag within the field-of-view of the camera at all times and to prevent loss of localisation, a piecewise linear trajectory between the current pose and the target pose to be followed by the tracking controller is generated. This yields a two step trajectory generated by Algorithm 1 to approach the target. The trajectory achieves the objective of maintaining the AprilTag within the camera field-of-view through first centring the target in the image frame by servoing in the y and z axes in the quadcopter body-frame B in Figure 1. Upon centring the tag within axial bounds set by ϵyz, the quadcopter approaches the tag at a constant velocity until a final distance to tag is reached. A linear quadratic regulator (LQR) is then used to track the piecewise linear trajectory. The overall structure of the controller combined with the reference trajectory generator is illustrated in Figure 3, where qm and ¨qIMU are defined as the pose estimate and acceleration states from quadcopter system, and qest and Ë™qest represent the filtered position and velocity. The tracking error between the reference state and the current state is defined as ¯x = x0 − x.”
Comment 4: There is no comparison of the data obtained by any quality criterion.
Response 4:
We have fully re-written Section 4.3 to address the lack of comparison between data sets. We now show a per-axis comparison between the reference trajectory, simulated quadcopter approach and real-world quadcopter approach to the AprilTag. Furthermore, we compare the datasets through qualitative measures of acceptance convergence time and maximum tracking error as described in Algorithm 1 on page 10, Section 4.3 on pages 16-17, and detailed in Figure 8 on page 17.

Reviewer 4 Report
Comments and Suggestions for Authors
It's a good work
Author Response
We are grateful for Reviewer 4 scores of our paper.
No comments provided by Reviewer 4.
Round 2
Reviewer 1 Report
Comments and Suggestions for Authors
The paper has been revised and improved in accordance with the modification requirements, and it is recommended to be accepted.
Reviewer 2 Report
Comments and Suggestions for Authors
Most of my concerns have been addressed, and I appreciate additional figures/ the appendix that shows the experiment setups.
Reviewer 3 Report
Comments and Suggestions for Authors
The authors have taken all my comments into account. No further comments.